Phenotypic variation in dorsal fin morphology of coastal bottlenose dolphins (Tursiops truncatus) off Mexico

Morteo Eduardo emorteo@uv.mx eduardo.morteo@gmail.com 1
Rocha-Olivares Axayácatl 2
Morteo Rodrigo 3
Weller David W. 4
1 Instituto de Ciencias Marinas y Pesquerías/Instituto de Investigaciones Biológicas, Universidad Veracruzana , Boca del Río/Xalapa , Veracruz , Mexico
2 Centro de Investigación Científica y de Educación Superior de Ensenada , Ensenada , Baja California , Mexico
3 Facultad de Ciencias Administrativas y Sociales, Universidad Autónoma de Baja California , Ensenada , Baja California , Mexico
4 Southwest Fisheries Science Center, Marine Mammal and Turtle Division, National Oceanic and Atmospheric Administration , La Jolla , CA , United States of America
Gandini Patricia
Electronic publication date: 2017 Jun 13
Publication date: 2017
Volume: 5
Electronic Location ID: e3415
Received 2017 Mar 4; Accepted 2017 May 14
Copyright: ©2017 Morteo et al.
Copyright year: 2017
Copyright holder: Morteo et al.
License: This is an open access article distributed under the terms of the Creative Commons Attribution License, which permits unrestricted use, distribution, reproduction and adaptation in any medium and for any purpose provided that it is properly attributed. For attribution, the original author(s), title, publication source (PeerJ) and either DOI or URL of the article must be cited.
License URL: https://creativecommons.org/licenses/by/4.0/

Keywords: Adaptations, Stepping stone model, Population discrimination, Polymorphism

Funding: SNI (Sistema Nacional de Investigadores) Department of Marine Ecology at CICESE Sergio Flores at UABCS Drew Talley at UC-Davis This research was part of the M.Sc. thesis of the lead author who benefited from a CONACyT (Consejo Nacional de Ciencia y Tecnología) graduate fellowship. Financial aid was also provided by a SNI (Sistema Nacional de Investigadores) scholarship provided by Saúl Álvarez-Borrego, and a thesis grant issued by the Department of Marine Ecology at CICESE. Logistics for the Gulf of California surveys were partially funded by Sergio Flores at UABCS and Drew Talley at UC-Davis. There was no additional external funding received for this study. The funders had no role in study design, data collection and analysis, decision to publish, or preparation of the manuscript.

==============================
Geographic variation in external morphology is thought to reflect an interplay between genotype and the environment. Morphological variation has been well-described for a number of cetacean species, including the bottlenose dolphin (Tursiops truncatus). In this study we analyzed dorsal fin morphometric variation in coastal bottlenose dolphins to search for geographic patterns at different spatial scales. A total of 533 dorsal fin images from 19 available photo-identification catalogs across the three Mexican oceanic regions (Pacific Ocean n = 6, Gulf of California n = 6 and, Gulf of Mexico n = 7) were used in the analysis. Eleven fin shape measurements were analyzed to evaluate fin polymorphism through multivariate tests. Principal Component Analysis on log-transformed standardized ratios explained 94% of the variance. Canonical Discriminant Function Analysis on factor scores showed separation among most study areas (p < 0.05) with exception of the Gulf of Mexico where a strong morphometric cline was found. Possible explanations for the observed differences are related to environmental, biological and evolutionary processes. Shape distinction between dorsal fins from the Pacific and those from the Gulf of California were consistent with previously reported differences in skull morphometrics and genetics. Although the functional advantages of dorsal fin shape remains to be assessed, it is not unlikely that over a wide range of environments, fin shape may represent a trade-off among thermoregulatory capacity, hydrodynamic performance and the swimming/hunting behavior of the species.

Introduction

Fin shape in aquatic organisms has been suggested to reflect unique anatomical and physiological adaptations to different environmental conditions (Aleyev, 1977; Pauly & Palomares, 1989; Fish, 1998; Weller, 1998; Wright, 2000), and this is also widely accepted in cetaceans (Fish & Hui, 1991; Berta & Sumich, 1999; Fish & Rohr, 1999; Reynolds, Wells & Eide, 2000; Morteo, 2003). Morphological variation of the dorsal fin, to some extent, has been used for population and/or species identification (Lang & Pryor, 1966; Aleyev, 1977; Fish, 1998; Weller, 1998; Morteo, Morteo & Rocha-Olivares, 2005; Felix et al., 2017).

The dorsal fin of delphinids is important at two functional levels: thermoregulatory and hydrodynamic. Little empirical evidence exists, however, regarding the integrated performance of dorsal fins for most cetacean species (Lang, 1966; Weller, 1998; Fish & Rohr, 1999; Meagher et al., 2002; Westgate et al., 2007; Barbieri et al., 2010; Pavlov & Rashad, 2012; Van der Hoop et al., 2014). Estimating integrated performance is challenging since plasticity may be in part regulated by the energetic cost of different swimming behaviors related to locating, chasing, handling, and ingesting prey, thus maneuvering abilities may be important in feeding success, and the dorsal fin may play an important role for swimming stabilization (Weller, 1998; Fish & Rohr, 1999). Also, the dorsal fin is the only appendage that is constantly exposed to ambient air, and thus is subject to different thermoregulatory conditions from the rest of the body (Meagher et al., 2002; Westgate et al., 2007; Barbieri et al., 2010).

Bottlenose dolphins (Tursiops truncatus) have a worldwide distribution, occupying a variety of ecological conditions, and show substantial intraspecific phenotypic variation (Walker, 1981; Vidal, 1993; Gao, Zhou & Wang, 1995; Goodwin et al., 1996; Hoelzel, Potter & Best, 1998; Turner & Worthy, 2003; Weller, 1998). Polyphenisms in traits whose functions arose as adaptations to new life conditions (e.g., aquatic for terrestrial ancestors) may be directly linked to the environment, and morphometric variations should be studied as a function of ecological differences (Stearns, 1989; Gotthard & Nylin, 1995). Here we analyze phenotypic variation of bottlenose dolphin dorsal fins in relation to respective habitats, ecology and behavior over different spatial scales. We evaluated the degree of fin polymorphisms of 19 putative populations from Mexico, contrasting them by location and region, in the context of relevant biological, ecological and geological features. The goal of the study was to determine if observed morphometric variations follow the stepping stone model, where the degree of differentiation among neighboring populations is correlated with the migration distance travelled by individuals (Wright, 1943; Kimura, 1953).

Figure 1 Study areas.

(1) Pacific Ocean: EN, Ensenada, Baja California; SQ, San Quintin, Baja California; BM, Bahia Magdalena, Baja California; MZ, Mazatlán, Sinaloa; BB, Bahia Banderas, Jalisco; PE, Puerto Escondido, Oaxaca; (2) Gulf of California: UG, Upper Gulf of California, Sonora; SJ, Bahia San Jorge, Sonora; BL, Bahia de los Angeles, Baja California; BK, Bahia Kino, Sonora; SM, Bahia Santa Maria, Sinaloa; LP, La Paz, Baja California Sur; (3) Gulf of Mexico: TA, Tamiahua, Veracruz; NA, Nautla, Veracruz; VR, Veracruz Reef System, Veracruz; AL, Alvarado, Veracruz; TB, Tabasco, Tabasco; TL, Terminos Lagoon, Campeche; HO, Holbox, Quintana Roo.

Methods

Study area

Sampling locations were selected considering the following: (1) Geographic coverage should include most of the species distribution within Mexican coastal waters, (2) Locations should represent most of the existing conditions of habitat variability for the species in Mexico, (3) Distances among adjacent locations should allow for individual exchange considering the dispersal capabilities of the species, and (4) Photo-identification catalogs of coastal bottlenose dolphin populations must be available. Detailed descriptions on the ecology of the study areas and the biology of dolphin populations in those areas are provided elsewhere (see Espinosa, 1986; Ballance, 1987; Salinas & Bourillón, 1988; Acevedo, 1989; Ballance, Leatherwood & Reeves (1990); Ballance, 1992; Delgado, 1996; Delgado, 2002; Caldwell, 1992; Heckel, 1992; Schramm, 1993; Silber et al., 1994; Silber & Fertl, 1995; López, 1996; López, 2002; Defran et al., 1999; Díaz, 2001; Orozco, 2001; Reza, 2001; Guzón, 2002; Morteo, 2002; Rodríguez, Lugo & Foubert, 2003; Ladrón de Guevara & Heckel, 2004; Morteo et al., 2004; Ramírez, Morteo & Portilla-Ochoa, 2005; Mellink-Bijtel & Orozco-Meyer, 2006; Pérez-Cortes, 2006; Rodríguez-Vázquez, 2008; Morteo et al., 2012; Morteo, Rocha-Olivares & Abarca-Arenas, 2014; Morteo et al., 2015; Morteo, Rocha-Olivares & Abarca-Arenas, 2017; Ruíz-Hernández, 2014; Zepeda-Borja, 2017, unpublished data). Study areas were grouped by region into (1) Pacific Ocean, (2) Gulf of California and (3) Gulf of Mexico (Fig. 1). For instance, (1) the Mexican Pacific (i.e., localities EN, SQ, BM, MZ, BB and PE in Fig. 1) features an open habitat with a narrow continental shelf as a result of active processes of plate tectonics, thus coastal bathimetry has a steeper slope (usually depths >40 m are reached at <2 km from the shore), where swells are typically high (>1.5 m); the average sea surface temperature (SST) turns warmer through a north-south gradient (15−30 °C) and productivity is mostly dominated by coastal upwellings via ocean circulation and local primary producers (i.e., kelp beds); also, except for the southern portion of the study area (i.e., locality PE in Fig. 1), rainfall and coastal vegetation have little influence on the ecology of these areas, even within the estruaries and lagoons. (2) Conversely, the Gulf of California is a semiclosed habitat where ocean currents are complex due to the intricate bathimetry and the tidal regime; it has an exceptionally high primary productivity driven mostly by seasonal upwellings, shallow thermoclines and a wind-mixed water column. The Gulf of California has been divided into three oceanographic and biogeographically different regions from north to south, such that: (a) northern coastal waters (i.e., UG and SJ in Fig. 1) are shallow (<10 m), usually warmer (>20 °C) with high salinity and strong tidal currents (up to 1 m s−1); in contrast b) the central coast (i.e., BL and BK in Fig. 1) is steeper due to the deep Canal de Ballenas and Tiburon Island passages (>1000 m), with colder SST (<20 °C) due to frequent upwellings, and features high swells (>2 m) formed by strong winds (>5 m s−1); and c) the southern area (i.e., LP and SM in Fig. 1) has shallow bays (<20 m) bordered by a deeper coastal waters (>100 m) situated at the entrance to the Gulf; wave height and SST are highly variable throughout the year (except in location SM) and so is primary productivity due to the influence of the several water masses coming in from the Pacific. Finally, the Gulf of Mexico (i.e., TA, NA, VR, AL, TB, TL and HO in Fig. 1) is a very shallow area (usually depths around 20 m are reached over 4 km from the shore) where tides are very low (<1 m) and most of the oceanic circulation is driven by the loop current that carries warm waters (mean SST >26 °C) from the Caribbean into the Gulf. Although the region is classified as an open habitat, many dolphin populations inhabit shallow (depth <10 m) lagoons (i.e., TA, TB and TL) or semi-protected coastal waters surrounded by reefs (i.e., VR) or islands (i.e., HO), thus swells are also very low (<1 m). Coastal productivity is usually higher around continental water bodies due to nutrient runoffs, especially during the rainy season; thus the ecology of most of these areas is strongly influenced by temporal changes in wind and rain regimes.

Photographic procedures

Dorsal fin shapes were obtained from high quality images; since our methods involved only non-invasive data collection (i.e., pictures were taken onboard a boat that was 15–50 m away from the animals), an institutional review board was unnecessary. Also, original photographs from wild dolphins were obtained through a federal permit (SGPA/DGVS/518) from Secretaría del Medio Ambiente y Recursos Naturales (SEMARNAT). The remaining images came from photo-identification catalogs in other published and unpublished scientific research; thus it was assumed that all these were approved by their institutional review boards (if applicable) and were issued with the federal permits for their field work, such that these can be consulted in each case. Most of the pictures were obtained during the late 90’s and the following decade, comprising at least 21 different years of information (see Table 1). The oldest photographic material was collected in the early 80’s (e.g., Bahía Kino by Ballance, 1987) or 90’s (e.g., Tamiaha Lagoon by Heckel, 1992; Schramm, 1993), but some catalogs were updated over the following years (e.g., Ensenada by Espinosa, 1986; Guzón, 2002); however, the average duration of sampling effort for each of these studies was 2.6 years (s.d. = 2.1) (see Table 1).

Table 1 Summary of data sources and sample size.

Abbreviations for study areas follow those in Fig. 1.

Source(s)	Area	Duration (y)	NCat	Sample (%)	Format	Pods	
Espinosa (1986), Defran et al. (1999), Guzón (2002)	EN	3	144	27 (19%)	S, T	20	
Caldwell (1992), Morteo (2002), Morteo et al. (2004)	SQ	2	220	29 (13%)	S, T	16	
Pérez-Cortes (2006)	BM	5	211	30 (14%)	S	27	
Zepeda-Borja (2017, unpublished data)	MZ	3	210	30 (14%)	D	26	
Rodríguez, Lugo & Foubert (2003), Rodríguez-Vázquez (2008)	BB	6	60	28 (46%)	D	12	
This work	PE	<1	24	21 (87%)	D	4	
This work	UG	<1	28	23 (82%)	D	3	
Orozco (2001), Mellink-Bijtel & Orozco-Meyer (2006)	SJ	1	217	24 (11%)	S, T	N.A.	
Ladrón de Guevara & Heckel (2004)	BL	1	26	19 (73%)	D, S, P, T	4	
Ballance (1987), Ballance (1990), Ballance (1992)	BK	2	155	30 (19%)	S, T	17	
Reza (2001)	SM	1	637	25 (4%)	S	N.A.	
Díaz (2001)	LP	1	66	29 (44%)	P	N.A.	
Heckel (1992), Schramm (1993)	TA	3	51	20 (39%)	S	N.A.	
Ramírez, Morteo & Portilla-Ochoa (2005)	NA	1	148	30 (20%)	S	17	
Ruíz-Hernández (2014), Morteo et al. (2015)	VR	2	93	30 (32%)	D	30	
Morteo et al. (2012), Morteo, Rocha-Olivares & Abarca-Arenas (2014), Morteo, Rocha-Olivares & Abarca-Arenas (2017)	AL	8	282	30 (11%)	S, D	30	
López (1996), López (2002)	TB	2	750	35 (4%)	S	28	
Delgado (2002)	TL	5	1,987	37 (2%)	D, T	N.A.	
Delgado (1996), Delgado (2002)	HO	3	344	36 (10%)	T	N.A.	
	Total	21a	5,653	533 (32.3%b)	–	234	
Notes.

NCat, Number of individuals in the catalog. Image format is classified by reliability from digital pictures (D), digitized negatives or slides (S), scanned pictures (P) and scanned traces (T). N.A., not available.

a Total number of different years.

b Weighted average.

Image quality was crucial for the analysis, thus the best image from each individual was selected from the photo-identification databases according to the following criteria (modified from Weller, 1998): (1) Images only of mature dolphins; (2) Dorsal fins entirely visible, as complete as possible, and non-parallaxed; (3) Fins size at least one ninth of the entire picture; (4) Whenever possible, pictures from individuals sighted in different schools were selected in order to minimize chances of genetic relatedness (i.e., trait heredity). Images not fulfilling at least the first three criteria were excluded. Due to the variety of sources and formats, 32% of the material came from film-based images, and a similar proportion was from digital pictures, whereas 28% were fin contour traces in paper and 8% came from printed pictures (see Table 1).

Approximately 30 different individuals were randomly selected from each locality; these were later compared to avoid potential inter-study area matches (which did not occur). All individuals were assumed to belong to the coastal form of the species, as specified in the original catalogs.

Digital measurements

We developed a software routine (Fin Shape v1.3) in the computer language Borland Builder C++ 5.0, to specifically measure angles and distances between landmarks of dorsal fins following Weller (1998) and Morteo, Morteo & Rocha-Olivares (2005). Images were digitized at high resolution (3,000 dpi), as needed, and measured consistently by a single trained operator (E. Morteo). Following Weller (1998), the anterior insertion point of the dorsal fin on the body (B) was identified by an abrupt change in the contour of the dolphin’s back; also, the tip of the dorsal fin (A) was identified as the landmark furthest from point B (Fig. 2). Once these two points were identified, a connecting straight line was automatically drawn, and additional lines were projected departing from B at 30°, 20°, 10° and 5° below segment AB. The operator then identified where these lines intersected the edge of the fin, and their lengths were computed (in pixels).

Figure 2 Reference points and measurements computed by FinShape software to acquire morphological landmarks.

A = Tip; B = Base. Points C5, C10, C20 and C30 indicate angles (degrees) relative to line AB. Point D is the intersection of a line departing from C30 with the fin’s leading edge, this line is also perpendicular to line AB. Point O is the intersection of lines AB and C30D. Surface area (shaded) was calculated considering line C30B as the limit.

Eleven measurements were obtained from each image along with the surface area (Fig. 2); these describe four qualitative shape parameters according to Weller (1998): (1) Base length: Distance from the leading edge to the trailing edge of the fin; (2) Depth: Distance from the fin tip to the anterior insertion on the body; (3) Rake: Amount that the tip of the fin extends beyond the base of the trailing edge; and (4) Foil: Curvature of the leading edge of the fin.

Since preliminary software trials by Morteo, Morteo & Rocha-Olivares (2005) showed that repeated measures of the same image, and also of several different images from the same individual, yielded very little variations (i.e., <0.1%), it was assumed that the operator was able to correctly identify the features of the fin, and that image quality across all photographic formats was sufficient to prevent measuring bias. Measurements were used to calculate 11 indexes for each individual through the following standardized ratios (modified from Weller, 1998): C30B/AB, C20B/AB, C10B/AB, C5B/AB, C30D/AB, C20D/AB, C10D/AB, C5D/AB, AO/OB, DO/C30O, AC302/area.

Morphological variation

Average dorsal fin contours were constructed by locality based on median adimensional ratio values. In order to scale these contours, segment AB was fixed to 10 relative units, thus all fin representations would have the same depth. The remaining segments were calculated through mathematical and trigonometrical equations that solved the related ratios using their correspondent median values (Morteo, Morteo & Rocha-Olivares, 2005) (see Appendix S1); since the latter were not normally distributed, point Cartesian coordinates were computed from each median measurement, and data dispersion was represented as quartiles (upper = 75% and lower = 25%) indicated by bars and ellipses around the calculated median points. This procedure follows a Procrustean approach (Peres-Neto & Jackson, 2001), where distortion, scale and rotation are controlled to provide variation specificity for each reference point measure within the fin.

Statistical analyses

Phenotypic variability was assessed on log-transformed ratios (Zar, 1996) at three geographic scales: (1) within localities, (2) among localities (Isolation-Differentiation by distance), and (3) among oceanic regions.

Variation within localities was designed as a test for sample representativity through a rarefaction analysis. Local coefficients of variation (COV) were computed for each log-transformed ratio, which were later averaged to obtain a general index for the local variability of the fin (GIV). Variation among localities and regions was assessed through multivariate analyses of log-transformed ratios; since most of these ratios were highly correlated, a Principal Component Analysis (PCA) was performed. Individual scores from the PCA were used to perform a Discriminant Function Analysis (DFA) (tolerance = 0.01) (Kachigan, 1991; Manly, 1994; Grimm & Yarnold, 1995). Assumptions for multivariate tests were verified, and a non-stepwise Canonical Discriminant Analysis (CDA) was performed to determine separation among samples (tolerance = 0.01). Also, p values in multiple comparisons were later tested for significant differences by using the sequential Bonferroni correction (Rice, 1989). All data were analyzed using Statistica v6.0 (Stat Soft™).

Finally, Squared Mahalanobis Distances (SMD) from the DFA were used to construct a dendrogram (Single Linkage Cluster Analysis) and dissimilarities were expressed as percentages (100*linkage SMD/Max SMD). SMDs were also used to perform an analysis of differentiation by distance, in which we correlated the matrix of phenotypic differentiation among localities (SMDs) with a matrix of geographic distances using a Mantel one-tailed test (α = 0.05, Monte Carlo and 10,000 permutations) as implemented in the Excel (Microsoft Office XP™) add-in XLStat-Pro v7.0 (Addinsoft™). Due to the coastal nature of these dolphins, geographic distances among localities (km) were calculated roughly following the coastline, thus these represent minimum separations among localities. Finally, we performed partial Mantel tests for each region in order to determine the contribution of each dataset to the general model.

Results

Data overview

A total of 5,653 dorsal fins were analyzed from the photo-id catalogs available at the 19 locations (Table 1). From all the images that fulfilled the quality criteria, a total of 533 individuals were randomly selected (representing 32.3% of fins or individuals from all the catalogs). Except for Bahia de los Angeles, the Upper Gulf of California, and Puerto Escondido, the fins used in this study accounted for less than half the number of identified individuals; also, when sighting data were available, for any particular location most fins came from different pods, such that the average proportion of individuals from different pods at each study area was 61.2% (s.d. = 12.8%).

General phenotypic variability

The Cartesian position of each landmark in the standardized fins varied in decreasing order as follows A→C30 → O → D → C5; also, landmarks C10 and C20 were the least variable in all cases (represented by smaller ellipses) (Fig. 3). Fins within the Gulf of Mexico (Fig. 3C) showed the least variability overall; these were also the least falcate, contrasting with all the fin contours from the Pacific (Fig. 3A) and most from the Gulf of California (Fig. 3B). Average fin contours featured a larger base length for the fins from the Pacific and the Gulf of California (except Bahia de los Angeles). Wide rounded tips were also found for most Pacific dolphins (except for Puerto Escondido), and pointed tips prevailed in dolphins from the northern Gulf of California (Fig. 3B) (except San Joge, Santa María and La Paz) and the Gulf of Mexico (Fig. 3C). Also, fins from the Pacific and the Gulf of California had more foil, and most of their tips did not extend further from the posterior basal landmark (i.e., less rake) as in fins from the Gulf of Mexico (except for Bahia de los Angeles and Bahia Kino in the northern Gulf of California). Finally, fins from the Gulf of Mexico and the northern Gulf of California were slightly taller (AC30), thus the surface area was also larger.

Figure 3 Median dorsal fin contours by study areas ( N = 533 dorsal fins).

Regional divisions are grouped in columns: (A) Pacific Ocean, (B) Gulf of California and (C) Gulf of Mexico. Contours reflect median values of shape and do not represent any particular dorsal fin. Study area codes follow those in Fig. 1, and sample size is shown in parenthesis; error bars and ellipses show variability expressed as quartiles (50% of data). Measurement AB (i.e., from the tip to the anterior insertion into the body) is the same for all fins (10 relative units).

Figure 4 Dorsal fin morphological variability and sample size effect within the 19 study areas (N = 533 dorsal fins).

Figure 5 Dendrogram based on morphometric distances among study areas (N = 533 dorsal fins).

Study area codes follow those in Fig. 1. Values are proportions based on the maximum Squared Mahalanobis distance (Table 3). Major branches are somewhat consistent with the regional division of the study areas: (1) Pacific Ocean (bold lines), (2) Gulf of California (dashed lines), (3) Gulf of Mexico (thin lines).

Variability within localities

As expected, variability within localities increased as more fins were included in rarefaction curves; however, these reached an asymptote at the 19th sample (>95% of the local variability) in most localities; therefore the minimum sample size was inferred as 20 individuals (Fig. 4).

Variability among localities

The PCA performed on all eleven log-transformed ratios showed that 94.7% of the variance was explained by the first three factors, and the remaining seven factors accounted for less than 2% each, thus they were not useful in the following analyses (Table 2).

Table 2 Factor Analysis of log-transformed ratios using all locations (N = 533).

Note that the cumulative variance accounted for the first three Factors (*). Discrimination was highly significant among the 12 locations (Wilks’ Lambda: 0.44442, F(33,922) = 8.8617, p < 0.00001, N = 533).

Factor	Eigenvalue	Cumulative variance (%)	Wilks’ Lambda	Partial Lambda	F-remove (11,313)	p-level	
1	5.69	44.89	0.70	0.61	18.90	<0.001	
2	2.98	72.79	0.52	0.81	6.76	<0.001	
3	2.43	94.74*	0.51	0.84	5.66	<0.001	

Scores from PCA were normally distributed (p > 0.2), and the DFA among the 19 locations was highly significant (Wilks’ Lambda: 0.44442, F(33,922) = 8.8617, p < 0.00001, n = 533). All three factors contributed significantly (p < 0.00001) to the model (Table 2).

SMDs (Table 3) revealed widespread differentiation among locations. All sites were significantly different (p < 0.01) to at least 12 other study areas (i.e., Bahia Magdalena, Mazatlan and Tamiahua). The most distinctive samples were from La Paz and Puerto Escondido (18 significant differences), followed by San Jorge, Bahia de los Angeles (16 each), and Santa Maria (15); all other locations were significantly different to 13 other sites.

Cluster analysis revealed two geographic groups, one corresponding to the localities from the Gulf of Mexico, including Santa Maria (Gulf of California) as a geographic outlier, and the second including localities from the Pacific and the Gulf of California exclusively (Fig. 5). Puerto Escondido was the most distinct locality and was not nested in any of the geographic groups.

A significant correlation was found between the Squared Mahalanobis and geographic distances (Mantel test, r = 0.35, p < 0.001); thus supporting the hypothesis of differentiation by distance and a stepping-stone dispersal model. Most of the contribution to the model came from samples within the Gulf of Mexico (partial Mantel test, r = 0.60, p < 0.001), followed by the Gulf of California (partial Mantel test, r = 0.50, p = 0.054) and the Pacific (partial Mantel test, r = 0.44, p = 0.093). This correlation broke down when samples from the Pacific and the Gulf of California were analyzed together (Mantel test, r = 0.20, p > 0.11).

Table 3 Matrix of squared Mahalanobis and geographic distances (N = 533).

Values over the diagonal are geographic distances along the coastline (km) and under the diagonal are Squared Mahalanobis distances. Location codes follow those in Fig. 1. Comparisons within regions are shaded.

Study area	EN	SQ	BM	MZ	BB	PE	UG	SJ	BL	BK	SM	LP	TA	NA	VR	AL	TB	TL	HO	
EN	0	220	1352	4140	4540	5748	2870	3040	2520	3340	3730	1820	–	–	–	–	–	–	–	
SQ	0.51	0	1132	3920	4320	5528	2650	2820	2300	3120	3510	1600	–	–	–	–	–	–	–	
BM	0.59	0.47	0	2788	3188	4396	1607	1737	1257	2007	2677	557	–	–	–	–	–	–	–	
MZ	0.41	0.34	0.60	0	400	1608	1439	1269	1789	969	279	2489	–	–	–	–	–	–	–	
BB	2.22*	0.59	0.58	0.62	0	1208	1720	1580	2020	1380	600	2720	–	–	–	–	–	–	–	
PE	3.80*	4.01*	3.74*	1.831*	2.28*	0	3047	2877	3397	2577	1887	3978	–	–	–	–	–	–	–	
UG	0.55	0.48	1.413*	0.87	1.67*	4.21*	0	170	350	450	1110	1050	–	–	–	–	–	–	–	
SJ	3.10*	1.18*	0.944	1.17*	0.18	5.03*	2.69*	0	480	300	980	1180	–	–	–	–	–	–	–	
BL	1.56*	1.30*	1.928*	0.96	2.40*	3.12*	0.35	3.55*	0	750	1420	700	–	–	–	–	–	–	–	
BK	1.06*	0.23	0.481	0.91	0.73	3.73*	0.71	1.10*	1.07	0	690	1450	–	–	–	–	–	–	–	
SM	4.60*	2.14*	1.817*	2.16*	0.75	4.68*	2.87*	0.85	2.88*	1.87*	0	2120	–	–	–	–	–	–	–	
LP	0.620	1.00*	0.898	1.92*	2.16*	5.32*	1.14	3.40*	2.72*	2.19*	4.536*	0	–	–	–	–	–	–	–	
TA	4.44*	2.24*	2.146*	1.71*	1.42*	1.57*	2.23*	1.77*	1.64*	1.66*	0.42	4.98*	0	137	284	348	695	786	1428	
NA	5.02*	4.65*	4.867*	3.19*	1.80*	1.31*	3.74*	2.18*	1.29*	2.18*	3.74*	2.68*	0.25	0	147	211	558	748	1326	
VR	4.34*	5.08*	5.321*	4.20*	2.12*	1.40*	2.91*	2.33*	1.19*	1.83*	4.19*	2.43*	0.34	0.29	0	64	411	601	1179	
AL	4.12*	3.77*	4.432*	4.07*	2.38*	1.32*	3.98*	2.48*	1.18*	1.39*	4.29*	2.60*	0.18	0.21	0.17	0	347	537	1115	
TB	4.29*	4.31*	4.876*	3.65*	2.46*	1.06*	3.65*	2.56*	1.32*	1.41*	3.59*	2.83*	0.47	0.40	0.37	0.28	0	126	768	
TL	4.34*	2.47*	4.145*	2.86*	2.04*	1.25*	1.89*	2.70*	1.17	1.98*	0.92	4.85*	0.15	0.52	0.35	0.47	0.21	0	642	
HO	6.14*	3.85*	3.143*	3.96*	3.07*	1.00	3.22*	3.75*	2.12*	3.32*	1.24*	6.52*	0.40	0.86	0.74	0.63	0.36	0.16	0	
Notes.

* significant distances (α < 0.01).

– Not applicable.

Discussion

Heredity and individual dispersal as mechanisms for phenotypic variability

Morphological variations in dorsal fin characteristics of bottlenose dolphins inhabiting Mexican coastal waters were observed, even among study areas in near proximity (Figs. 1 and 5); this variation was evident in the averaged fin contour reconstructions (Fig. 3), which showed significant differences among most localities (Table 3, Fig. 5). Such statistical discrimination may be largely due to the low phenotypic variability within putative populations, such that a small number of individuals (∼20) were needed to obtain at least 95% of the expected variability within any given locality (Fig. 4). Therefore, it is reasonable to assume that dorsal fin shape may be under strong selection acting over individuals with particular dorsal fin phenotypes. However, methodological and biological contexts must be considered in order to prevent misleading interpretations. For instance, the use of photographs may introduce some level of unaccounted bias in age/sex representation in the sample (Hersh & Duffield, 1990; Gao, Zhou & Wang, 1995; Weller, 1998; Perrin & Mesnick, 2003).

A similar line of reasoning can be applied to the regional clusters found in this study. For instance, the high migratory rates documented for bottlenose dolphins in the northern Mexican Pacific (Defran et al., 2015) may explain the inter-study area dorsal fin similarities (Fig. 3) and low dissimilarity values (Table 3) despite the large distances among sites (Figs. 1 and 5). Detailed body morphometrics provided by Walker (1981) already support phenotypic similarity of bottlenose dolphins along the west coast of Baja California.

Results from the Gulf of California stand in sharp contrast to those from the Pacific in that even when geographic separation among localities was relatively low, 73% of the comparisons within this region showed significant differences (Table 3, Fig. 5). There is currently no information on migration rates for dolphins among all of these areas; however, based on morphometric analyses of skulls from dead stranded bottlenose dolphins, Vidal (1993) suggested a geographic regionalization akin to the differentiation found in this study. This pattern has been reported in several other taxa including invertebrates (Correa & Carvacho, 1992; De la Rosa et al., 2000), fish (Riginos & Nachman, 2001), and other marine mammals like the California sea lion (Zalophus californianus californianus) (Schramm, 2002; Aurioles et al., 2004; Pedernera et al., 2004; Porras et al., 2004). Contrasting oceanographic patterns (Lavín, Palacios-Hernández & Cabrera, 2003) may contribute to this separation, causing habitat and resource partitioning. Moreover, Segura et al. (2006) also found genetic structures in bottlenose dolphins within the Gulf of California, which supports our findings. Overall, molecular and phenotyoic co-variation (i.e., skull and dorsal fin morphometrics) point to the possibility of the early steps of microevolutionary divergence in T. truncatus from the Gulf of California.

Conversely, no significant morphological differences were found within the Gulf of Mexico, but fin shapes were significantly correlated with distance among locations (Table 3, Fig. 5). Similar to the highly migratory movements of dolphins along the Pacific coast, bottlenose dolphins in the Gulf of Mexico also appear to have large home ranges. For instance, Delgado (2002) documented one individual that moved at least 800 km from Holbox Island to the western coast of the Gulf in less than a year; he also found four other individuals that moved 240–320 km in 274–1,404 days. Martínez-Serrano et al. (2011) and Morteo, Rocha-Olivares & Abarca-Arenas (2017) also found large home ranges (>100 km) and individual movements (100-300 km) for dolphins in the northwestern Gulf of Mexico. Since dolphins may potentially reproduce with individuals from other locations, genetic exchange occurs over a large scale (Islas, 2005); however, evidence also suggests a certain degree of genetic structure and restricted gene flow that is consistent with sex-specific dispersal patterns (Islas, 2005; Caballero et al., 2012), which may enhance homogeneity in character states by male dispersion, but also promote differentiation through female restricted home ranges (e.g., Morteo, Rocha-Olivares & Abarca-Arenas, 2014), potentially resulting in the observed clinal morphological trends.

Male dispersal patterns seem to be a dominant feature in the western Gulf of Mexico (Morteo, Rocha-Olivares & Abarca-Arenas, 2014), thus our morphological results seem concordant with genetic data; however, morphological similarities in this trait may be also be attributed to the homogeneity and stability of the environment (a possible mechanism is further discussed in ‘Could dorsal fin phenotypic variability be an adaptive trait?’). For instance, compared to the Gulf of California, tides, wave regimes and SST variations in the Gulf of Mexico are much lower in magnitude and frequency due to the influence of the Loop Current (Davis et al., 1998); and because of its influence over a large area (Avise, 1992), selective pressures may be similar in spite of the larger distance among sites. This situation may explain not only the similarities in shape among neighboring locations, but also the smaller overall variability observed in these dorsal fins (Fig. 3).

Dorsal fin phenotypic variability as a function of geographic/geological scales

Similar studies have also pointed out the potential utility of using geographic variation in dorsal fin shape for bottlenose dolphin population discrimination over wide geographic ranges (i.e., Indian and Pacific oceans vs. the Gulf of Mexico), thus intrinsic differences are somewhat implied (Weller, 1998); however, due to the wide geographic scale of such comparisons, random fin phenotypes may occur due to vicariance. Therefore, there was no other study to provide further insight in the dorsal fin morphology of bottlenose dolphins by using a larger sample at a smaller scale.

Moreover, the major differences in dorsal fin shapes found between the Pacific, the Gulf of Mexico and the Gulf of California were consistent with the stepping-stone dispersal model (Table 3); however, there was no clear difference between the Pacific and the Gulf of California.

The geological history of the region may shed light on the matter; for instance, the divergence between the Gulf of Mexico and Pacific populations ensued after the emergence of the Isthmus of Panama, which closed the inter-oceanic canal (approx. 2.5 Mya) (Gore, 2003); therefore, morphological differences with the Gulf of Mexico may also be attributed to vicariance. Conversely, the Baja California peninsula began to separate from the mainland about 5.5 Mya; during this separation (5.5–1 Mya) the peninsula was fragmented on several occasions by trans-peninsular seaways, connecting the Pacific and the Gulf (Riddle et al., 2002). It is unclear how these hypothetical connections between the proto-Gulf of California and the Pacific may have affected coastal populations of bottlenose dolphins; however, this intermittent isolation may account for the lack of differentiation in dorsal fin shape between the Pacific (Ensenada and San Quintin) and the northern Gulf of California.

Could dorsal fin phenotypic variability be an adaptive trait?

Local phenotypic variability in dorsal fins may be the result of individual heredity and dispersal, but natural selection may also be at play. While empirical data on the possible functional advantages of dorsal fin characteristics are sparse, it has been argued that some features are adaptive (Fish & Hui, 1991; Weller, 1998; Berta & Sumich, 1999; Fish & Rohr, 1999; Reynolds, Wells & Eide, 2000). As a whole, dorsal fin shape may be a trade-off between thermoregulatory capacities and hydrodynamic performance. For instance, regardless of the study location, the central portion of the trailing edge (represented by points C10 and C20, Fig. 2) was the least variable section of the dorsal fin overall (Fig. 3). This finding may reflect a hydrodynamic constraint, since computer simulations for hypothetical dorsal fins have found that this region produces the least turbulence (Pavlov & Rashad, 2012). On the other hand, studies on thermoregulation efficiency in dorsal fins point out the importance of surface area and vascularization in temperature regulation (Meagher et al., 2002; Morteo, 2004; Westgate et al., 2007; Barbieri et al., 2010). For instance, veins and vessels in the dorsal fins of male dolphins are directly connected to the testicles (which are inside the body), helping in heat dissipation (Rommel et al., 1994; Rommel, Pabst & McLellan, 1999).

The functional significance of other morphometric characters (i.e., foil, rake, depth, base lenght) are less clear. Weller (1998) explained how the shape of surfboard fins and water craft keels confer different levels of maneuverability, speed and performance; he further suggested how variation in these parameters in the dorsal fins may relate to site specific hydrodynamic performance required for prey chase and capture by dolphins. For instance, contrasting fin shapes have been described for the two ecotypes of this species in the southeast Pacific which have different feeding habits, such that offshore individuals generally have wider and more falcate fins than their coastal counterparts (Felix et al., 2017).

Alternatively, dorsal fin shape characteristics may be influenced by habitat variables unrelated to prey capture. For instance, falcate and wider fins (which theoretically perform better at high speeds or in highly dynamic environments) were found mainly in the Pacific (Fig. 3), where the habitat of coastal bottlenose dolphins features swells as high as 5 m (Lizárraga et al., 2003), and also in the Gulf of California, where tidal currents may exceed 3 m s−1 (Álvarez, 2001). Comparatively, less falcate fins were found in the Gulf of Mexico, where tidal currents and wave heights are of lesser magnitude (Delgado, 2002).

Additionally, dorsal fins that were less falcate and with less foil were found in the western Gulf of Mexico, whereas taller and wider (i.e., larger depth and base length) dorsal fins were found in the northern Gulf of California; both of these features resulted in larger fin surface areas. These coastal locations are very shallow (<20 m) and have a high potential for rising sea surface temperatures (SST) due to high residency times resulting from reduced water circulation, especially during low tides (Bianchi, Pennock & Twilley, 1999; Lavín & Marinone, 2003). SST plays an important role in species distributions (Pianka, 1994; Valiela, 1995), and although it is not supposed to influence dolphins movements overall (due to their high thermoregulatory capacities), tagged bottlenose dolphins in the Atlantic have shown avoidance of oceanic fronts (Wells et al., 1999). Consequently, rapid changes in temperature may trigger behavioral and physiological responses possibly influencing home ranges, and also food habits and consumption rates. Thus a possible cause for the apparent relationship with proportional dorsal fin area may be that warmer habitats are more suitable for individuals that are better at handling heat excess.

In light of the above, there is a chance that unexpected morphological similarities between geographically isolated localities (e.g., Santa Maria in the Gulf of California vs. Gulf of Mexico sites, Fig. 5) are not an artifact of the classification functions (Table 2), and may have a biological/adaptive explanation. For instance, unlike the other study areas in the Gulf of California, Bahia Santa Maria is an enclosed estuarine system with two entrances, and the vegetal coverage along the shore is dense (Reza, 2001). Also, mean year-round SSTs are more similar to those in the Gulf of Mexico than to the open Pacific coast (Heckel, 1992; Schramm, 1993; Delgado, 1996; Delgado, 2002). Therefore, we speculate that similar dorsal fin shapes in the Gulf of California and the Gulf of Mexico may reflect adaptive convergence influenced by similar selective pressures.

We acknowledge that the relations described above may be coincidental and the former arguments are exploratory. Therefore, independent evidence is needed to understand if these polymorphisms reflect adaptive advantages and genetic mechanisms within and among populations, or are just the result of different norms of reaction. Although the patterns of morphological variation are somewhat consistent with biological and ecological features, suggesting adaptive explanations for such differences, hydrodynamic and thermoregulatory functions must be empirically assessed to determine if the character states found in this study are different enough to influence individual fitness, and thus subject to selection.

Conclusions

Dorsal fins of bottlenose dolphins show a high degree of polymorphism and restricted local variability. Dorsal fin polymorphisms were geographically structured at different spatial scales, supporting the model of isolation/differentiation by distance overall. Genetic analyses may help elucidate if the population structure is consistent with the morphological clinal variation described here. Our findings also suggest that this trait may be influenced by natural selection, but this hypothesis remains to be tested.

Supplemental Information

Supplemental Information 1 Dataset

Raw digital measurements of dorsal fin samples from each study area.

Click here for additional data file.

Supplemental Information 2 Summary of statistical tests and graphs

Sheet (1) The Variation Coefficients computed to generate the General Index of Variability (GIV) for Fig. 4 in the manuscript.

Sheet (2) The normality tests along with the box & whiskers plot for the log transformed measurement ratios.

Sheet (3) The summary of the Factor analysis for Table 2 in the manuscript.

Sheet (4) The summary of the Discriminant function analysis for Table 2 in the manuscript.

Sheet (5) The input data for the Mantel’s tests for Table 3 in the manuscript.

Sheet (6) The result of the Cluster Analysis for Fig. 5 in the manuscript.

Sheet (7) The results of the Mantel’s test for all the study areas.

Sheet (8) The results of the partial Mantel’s test for site comparisons between the Pacific Ocean and the Gulf of California.

Sheet (9) The results of the partial Mantel’s test for site comparisons within the Pacific Ocean.

Sheet (10) The results of the partial Mantel’s test for site comparisons within the Gulf of California.

Sheet (11) The results of the partial Mantel’s test for site comparisons within the Gulf of Mexico.

Click here for additional data file.

Appendix S1 Mathematical relations among the morphometric indices of dorsal fin shapes

The procedure to reconstruct average dorsal fin contours from point measurements.

Click here for additional data file.

This research was part of the M.Sc. thesis of the lead author. We acknowledge the curators and owners of the photo-identification catalogs, which were used and analyzed under their supervision and permission (federal permits for their field work can be consulted in each case): Lisa Ballance, Martha Caldwell, RH Defran, Alberto Delgado-Estrella, Raúl Díaz-Gamboa, Oscar Guzón, Gisela Heckel, Paloma Ladrón de Guevara, Irelia López, Eduardo Lugo, Adriana Orozco, Héctor Pérez-Cortés, Tania Ramírez, Maru Rodríguez, Ivett Reza, Mario Salinas, and Yolanda Schramm.

Additional Information and Declarations

Competing Interests

Author Contributions

Animal Ethics

Field Study Permissions

Data Availability

The authors declare there are no competing interests.

Eduardo Morteo conceived and designed the experiments, performed the experiments, analyzed the data, contributed reagents/materials/analysis tools, wrote the paper, prepared figures and/or tables, reviewed drafts of the paper.

Axayácatl Rocha-Olivares and David W. Weller conceived and designed the experiments, contributed reagents/materials/analysis tools, wrote the paper, reviewed drafts of the paper.

Rodrigo Morteo performed the experiments, contributed reagents/materials/analysis tools, wrote the paper, reviewed drafts of the paper.

The following information was supplied relating to ethical approvals (i.e., approving body and any reference numbers):

Our methods involved only non-invasive data collection (i.e., images were taken onboard a boat that was 15–50 m away from the animals), thus a Review Board was unnecessary.

The following information was supplied relating to field study approvals (i.e., approving body and any reference numbers):

Original photographs from wild dolphins were obtained through federal permit from Secretaría del Medio Ambiente y Recursos Naturales (SEMARNAT) issued to a former thesis committee member (Gisela Heckel), with approval number: SGPA/DGVS/518. The remaining images came from photo-identification catalogs from other published and unpublished scientific research; thus it was assumed that all these were approved by their Institutional Review Boards (whenever this applied) and were issued with the federal permits for their field work, such that these can be consulted in each case.

The following information was supplied regarding data availability:

The dataset is provided in a Supplementary File with a detailed summary of the statistical tests and graphs. The procedure for the reconstruction of average dorsal fin contours is detailed in Appendix A of a technical report, which was also provided as a Supplementary File.

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
