# Peer review of "Phenotypic variation in dorsal fin morphology of coastal bottlenose dolphins (Tursiops truncatus) off Mexico"

_PeerJ, doi:10.7717/peerj.3415_

## Round 0.1 · original submission · Major Revisions

I agree with the reviewers that it is an interesting paper with a number of issues that should be clarified and corrected before publication.

When revising your MS, I sincerely ask you to adequately address EACH of the referees' comments, by either incorporating the suggestions in the revision, if possible, or providing brief but convincing rebuttals in case you do not agree with them. Both reviewers agree you have to check your references perhaps choosing only the needed ones. There are some minor corrections and suggestions made through the text, and also one of the reviewers questioned the inference made with the results. Please also check where the reviewers may have provided additional advice or suggestions.

Most importantly, you should pay particular attention to ALL MAJOR COMMENTS concerning severe flaws of your work, the proper addressing of which requires either profound revisions of your Ms (and possibly also further sampling and analysis work) or particularly well-founded rebuttals. When you submit the revised manuscript, please also provide a detailed RESPONSE. The response should be uploaded as a separate word file, in addition to the files containing the revised Ms itself.

·

Basic reporting

Overall very well-written and interesting manuscript, with good English. Some relatively minor typos or grammar errors (specific comments provided below).
Sufficient background and context provided. Relevant figures provided, and raw data shared.
Relevant results presented.

Specific comments are provided below.

Experimental design

A very interesting study that I enjoyed reading. I commend the authors for their work.
Questions are well defined, investigation appears sound and rigorous, methods described in sufficient detail, unless noted otherwise (see specific comments).

Note that I cannot assess the Appendix A, as it is in Spanish.

Validity of the findings

I think the results justify the conclusions, and the authors appear to have made good use of existing literature and knowledge to place their results in a wider context.

I do propose one additional potential biological explanation for some of the results (see specific comments below), which I encourage the authors to consider.

Some additional specific comments are provided below.

Additional comments

GENERAL REMARKS

I found this study really interesting and well-written. I commend the authors for their work.
Please find some specific suggestions or corrections below.



SPECIFIC COMMENTS

INTRODUCTION

L 71: “as a function” or “as functions” (rather than “as function”)


METHODS

L 84-85: For point number 2, should “within Mexican waters” be added here? In other words, do the authors mean as conditions of habitat variability for the species in general, or for the species WITHIN Mexico? There is a distinction between the two, with certain implications. Therefore, it would be useful if the authors can clarify that.

L 89-96: This is a whole lot of references and a lot of information that the reader needs to go dig on their own. I wonder whether some kind of a brief summary of the ecology of these sites and the biology of the dolphin populations could be given here. A few lines highlighting major differences would suffice. I think adding this information would be extremely useful for the context of the study.

L 85-87: I assume number 3) probably refers to adjacent locations (or at least those within the same ocean basin), rather than all of them, since two different ocean basins are considered here.

L 108 (entire section): If would be useful if this section included some information on period of sampling. I.e. the period in which photographs used in the study were collected, and how far apart (in time) they were.

L: 108 (entire section): Information is missing on whether these were digital photographs, film-based photographs, or both. There is some indication later on that some (all?) photographs were digitised, suggesting at least some pictures must have been film-based. But this should be clarified.

L 109-111: I think the sentence about institutional review board is not needed here. It is already provided in the Declarations when submitted.

L 120: Change to: “Images only OF mature dolphins” (rather than “from”)
L 120-121: There seems to be an inconsistency here: “fins entirely visible” seems to contradict “as complete as possible”. Which is it, entirely visible or as complete as possible? Also, what is meant by “complete”?

L 121-123: What exactly does this mean? That only a single individual from each school was used? Or that all individuals from each school were used? It is not clear what criterion exactly is enforced here. It would be helpful if this can be clarified.

Figure 2: I often wonder about this, as I had a similar consideration myself: how do you determine the exact location of B (base)? Since this area of the body is a sloping curve, how does one determine the exact baseline point? The choice of which will affect all the resulting calculations. Furthermore, how does one ensure that the choice of this point is consistently applied across all fins? Perhaps this can be further clarified?

Figure 2: The authors state: “Points C5, C10, C20 and C30 indicate angles (degrees) relative to line AB.”
OK, but how are C5, C10 and C20 determined? I understand that they represent angles, but what is the choice of the points themselves? It is clear to me for C30, but not for the rest. I assume they must somehow be fixed in order to calculate the angles.
On the other hand, if the angles are fixed and the points are the result of that, then the angle should be provided. Some part of information seems to be missing.

L 155: Shouldn’t it be “Procrustean” rather than “Procustean”?

L 167: Shouldn’t it be “multivariate” rather than “multivariated”?

L 168-169: “a Principal Component Factor Analysis (PCA) was performed”
Is this correct? From my understanding, PCA and FA are two different methods. Can you clarify what you mean by PCFA?

L 171: Shouldn’t it be “multivariate” rather than “multivariated”?

L 185: I suggest consistency, e.g. Mantel test (like used earlier) instead of Mantel’s. Whichever is chosen, use throughout the text.



RESULTS

L 196: Can you clarify what exactly this number represents?

L 214: Please clarify what is meant by “foil” (fin surface area?), since this has not been defined or referred to in the manuscript previously. This term is also used later on in the discussion, so it would be helpful for the reader to be clear on what it means.

Figure 3, L 222: Please clarify what is meant by “inner circles”, it is not entirely clear.

L 237-239: 3+7=10, but there were 11 ratios. The numbers don’t seem to add up.



DISCUSSION

L 307-310: I see the point here, but perhaps the sentence could be modified slightly: it starts off with citing Weller 1998 in explaining surfboard fins and watercraft keels, but then moves on to prey chase and capture. I understand the meaning, but in its current form this sentence reads as if surfboards and watercraft have shapes that enable prey capture... Therefore, a slight modification of the sentence is required.

L 313: Remove “are” between the words “swells” and “as”.

L 326: avoidance OF cold oceanic fronts

L 318-330: Very interesting and well-written paragraph, which seems to be well supported by evidence. However, I wonder whether there may be an additional possible explanation that could be added. Could a wider surface area help in manoeuvring capabilities of dolphins? In that case, a more coastal, shallow habitat, with potentially lots of reefs, islands, channels or shallows, may require dolphins to be more mobile or agile in water than their counterparts in open oceans. From what I remember, this is one distinguishing feature among oceanic and coastal Tursiops, where oceanic form has much more slender fins (both dorsal and pectoral). Furthermore, there is some evidence that dorsal fins may be less relevant in oceanic environments, where fine-scale manoeuvring may be less important, such as in right-whale dolphins (Lissodelphis).
Finally, river dolphins, which live in very shallow and complex habitats, have very wide pectoral fins, presumably to aid in manouvering – but on the other hand, they do mostly lack dorsal fins, so this does not necessarily help the case here.
The discussion in the next paragraph, on Bahia Santa Maria and dense coastal vegetation, may lend some support to this notion. I would encourage the authors to consider this as an alternative, or better yet, additional possibility acting in synergy.

L 372: Remove the comma after “skull”.

L 376: Add a comma after “found”.

L 380: The word “gulf” should probably be capitalized into Gulf, since it is a geographic reference.

L 381: Remove the word “between”.

L 398: Change “neigborging” to “neighboring”.

L 405: I suggest “geographic”, to be consistent with the rest of the manuscript.

L 407: “...at A smaller scale”

L 432: Change “termoregulatory” into “thermoregulatory”.




REFERENCES

Morteo et al. in press appears twice in the References.

Reviewer 2 ·

Basic reporting

This paper provides new information on the phenotypic variation of the dorsal fin for a cosmopolitan species of dolphins. The information is sound, and the analysis methods are correct. It uses an acceptable and unambiguous English. The information given during the introduction is correct and put the theme in context and perspective. Nevertheless the authors use a superabundance of references, with some redundant examples throughout the text. The length of the manuscript is acceptable and the tables and figures help to understand the analysis performed. The results are meaningful for the objectives stated, but the inferences made on the function of the fin as and adaptive character are far from being supported by the data. Discussion and conclusion on the phenotypic variability as a function of heredity, on the other hand are insightful and can be sustained by the data. The Appendix A is very useful to replicate the methods, but it is written in Spanish.

Experimental design

The research falls within the scope of the journal, and de question is relevant. Given the context in which this paper is set, a gap in the information is filled, but part of the context regarding the adaptive value of the phenotypic differences found are not directly supported by data. Methods are correct and well implemented, but I would give some more detail on the photographs selection procedure. Analytical methods are well explained and are reproducible.

Validity of the findings

The results are meaningful for the objectives stated, but the inferences made on the function of the fin as and adaptive character are far from being supported by the data. Discussion and conclusion on the phenotypic variability as a function of heredity, on the other hand are insightful and can be sustained by the data. Statistically the study is well executed, and the selected methods are appropriate for objectives. The conclusions regarding the differences in morphotypes and its utility to separate populations can be used as a base for the inference of its adaptive value. The speculation on the shape and surface on of the fin related to its value as a hydrodynamic related feature are highly speculative.

Additional comments

The paper is very interesting in terms of finding phenotipyc characters and may lay the basis for a research line on the adaptative value of the fin shapes.

---

## Round 0.2 · accepted · Accept

Thank you very much for submitting your revised manuscript to PeerJ. I have assessed your revisions and am happy to Accept the article.